# COVID-19 Vaccination Status as Well as Factors Associated with COVID-19 Vaccine Acceptance and Hesitancy among Prisoners and the Implications

**DOI:** 10.3390/vaccines11061081

**Published:** 2023-06-09

**Authors:** Alina Shabir, Noorah A. Alkubaisi, Amna Shafiq, Muhammad Salman, Mohamed A. Baraka, Zia Ul Mustafa, Yusra Habib Khan, Tauqeer Hussain Malhi, Johanna C. Meyer, Brian Godman

**Affiliations:** 1Department of Medicines, Tehsil Headquarter (THQ) Hospital, Dera Ghazi Khan 32200, Pakistan; alinashabir90@gmail.com (A.S.); amnashafiq573@gmail.com (A.S.); 2Department of Botany and Microbiology, College of Science, King Saud University, Riyadh 11451, Saudi Arabia; nalkubaisi@ksu.edu.sa; 3Institute of Pharmacy, Faculty of Pharmaceutical and Allied Health Sciences, Lahore College for Women University, Lahore 54000, Pakistan; msk5012@gmail.com; 4Clinical Pharmacy Program, College of Pharmacy, Al Ain University, AlAin Campus, Al Ain P.O. Box 64141, United Arab Emirates; 5Clinical Pharmacy Department, College of Pharmacy, Al-Azhar University, Cairo 11651, Egypt; 6Discipline of Clinical Pharmacy, School of Pharmaceutical Sciences, Universiti Sains Malaysia, Penang 11800, Malaysia; 7Department of Pharmacy Services, District Headquarter (DHQ) Hospital, Pakpattan 57400, Pakistan; 8Department of Clinical Pharmacy, College of Pharmacy, Jouf University, Sakaka 72388, Saudi Arabia; yhkhan@ju.edu.sa (Y.H.K.); thhussain@ju.edu.sa (T.H.M.); 9Department of Public Health Pharmacy and Management, School of Pharmacy, Sefako Makgatho Health Sciences University, Ga-Rankuwa 0208, South Africa; hannelie.meyer@smu.ac.za (J.C.M.); brian.godman@strath.ac.uk (B.G.); 10South African Vaccination and Immunisation Centre, Sefako Makgatho Health Sciences University, Ga-Rankuwa 0208, South Africa; 11Department of Pharmacoepidemiology, Strathclyde Institute of Pharmacy and Biomedical Science (SIPBS), University of Strathclyde, Glasgow G4 0RE, UK; 12Centre of Medical and Bio-Allied Health Sciences Research, Ajman University, Ajman P.O. Box 346, United Arab Emirates

**Keywords:** COVID-19, prisoners, vaccines, status, acceptance, hesitancy, Pakistan

## Abstract

Prisoners form a population who are highly vulnerable to COVID-19 due to overcrowding, limited movement, and a poor living environment. Consequently, there is a need to ascertain the status of COVID-19 vaccination and factors associated with hesitancy among prisoners. A cross-sectional questionnaire-based study was undertaken among prisoners at three district jails in Punjab Province, Pakistan. A total of 381 prisoners participated and none of the study participants had received an influenza vaccine this year. In total, 53% received at least one dose of a COVID-19 vaccine, with the majority having two doses. The top three reasons of vaccine acceptance were “fear of contracting SARS-CoV-2 infection” (56.9%), “desire to return to a pre-pandemic routine as soon as possible” (56.4%), and “having no doubts on the safety of COVID-19 vaccines” (39.6%). There was no statistically significant difference (*p* > 0.05) in any demographic variables between vaccinated and unvaccinated prisoners except for age, which was strongly association with COVID-19 vaccine uptake (χ^2^(3) = 76.645, *p* < 0.001, Cramer’s V = 0.457). Among the unvaccinated prisoners (*N* = 179), only 16 subsequently showed willingness to receive a COVID-19 vaccine. The top three reasons for hesitancy were: COVID-19 is not a real problem/disease (60.1%), safety concerns (51.1%), and COVID-19 vaccine is a conspiracy (50.3%). Efforts are needed to address their concerns given this population’s risks and high hesitancy rates, especially among younger prisoners.

## 1. Introduction

The coronavirus disease (COVID-19) has altered the dynamics of world health with more than 758 million positive cases and 6.85 million deaths throughout the world as of 28 February 2023 [1]. In Pakistan, the first positive case of COVID-19 was reported on 26 February 2020. Following this, a substantial number of positive cases of COVID-19 have been reported throughout the country in different disease waves [2,3]. In Pakistan, 1.58 million positive cases have been reported along with 30,643 deaths up to 20 March 2023 [4]. Currently, the country is in the sixth COVID-19 wave, and approximately 50 positive cases are being reported every day [5]. 

Since the emergence of SARS-CoV-2 in 2019, the scientific community has been looking for an effective treatment against COVID-19 in order to normalize the situation, with the only alternative being the introduction of public health measures. These included strict lockdown measures with their associated impact on the lives and income of citizens in low- and middle-income countries (LMICs), including Pakistan, increasing poverty levels and unemployment [6,7,8,9,10]. Overall, it was estimated that there would be a 33.7% increase in poverty levels in Pakistan as a result of lockdown and social distancing measures including restrictions on travel [11].

In the absence of effective repurposed treatments in patients with COVID-19 (apart from dexamethasone) despite the hype, there was a considerable global need for effective vaccines [12,13,14,15,16,17,18]. Subsequent rigorous testing by several pharmaceutical industries led to the development of various vaccines against COVID-19 [19]. The mRNA-based and adenovector vaccines demonstrated satisfactory outcomes in clinical trials; consequently, health authorities around the globe were looking to initiate mass vaccination campaigns to vaccinate their populations as soon as possible [19,20,21,22,23,24]. This reflected the fact that, in order to avoid continued high morbidity, mortality, and economic loss associated with COVID-19, along with the high burden on the healthcare delivery system and personnel caused by the COVID-19 pandemic [25,26,27], the majority of the country’s population need to acquire immunity through a comprehensive vaccination campaign [28].

Pakistan has been fortunate to have a variety of COVID-19 vaccines available at COVID-19 vaccination centers for public use. Currently, eight COVID-19 vaccines are available for administration in Pakistan. These include the BBIBP-CorV vaccine (Sinopharm), CoronaVac vaccine (Sinovac), Ad5-nCoV vaccine (CanSino), mRNA-1273 vaccine (Moderna), ChAdOx1-S vaccine (AstraZeneca), BNT162b2 vaccine (Pfizer-BioNTech), Gam-COVID-Vac vaccine (Sputnik V), and the Pakistani-made Pakvac vaccines, which have been approved by the Drug Regulatory Authority of Pakistan (DRAP) for use in the COVID-19 national immunization campaigns [28,29].

Similar to other countries, the COVID-19 vaccination drive in Pakistan started in June 2020 and in the first phase, front line healthcare workers (HCWs) were provided with the vaccine free of charge [30]. In the next phase, the older population were vaccinated with the establishment of multiple COVID-19 vaccine centers in every city. Alongside this, there have been rigorous COVID-19 vaccination campaigns in Pakistan bolstered by the vaccine being provided free of charge; otherwise, the costs would have been prohibitive for an appreciable number of citizens in the country [31,32]. The mass-vaccination campaigns were backed up by print and electronic media [33]. This, along with establishing COVID-19 vaccine centers across the country, mobile COVID-19 vaccine services throughout Pakistan, and door-to-door campaigns, resulted in an estimated 75.5% of the population becoming fully vaccinated and an estimated 78.0% of the population receiving at least one dose of a vaccine by the beginning of March 2023 [30,34].

Prisoners are considered to be at a high-risk of being infected with COVID-19 due to a number of factors. These include over-crowding and restricted movement in prisons [35,36]. Overall, more than two million prisoners globally are highly vulnerable to COVID-19 due to poor health and living conditions [37]. The influx of people from outside into jails is constant, including security personnel, cleaners, lawyers, and family members, adding to the risk of infection. Consequently, once a single prisoner is infected, an outbreak of severe acute respiratory syndrome coronavirus 2 (SARSCoV-2) could rapidly develop, increasing morbidity and mortality [38]. Multiple large outbreaks of COVID-19 have been reported in prisons and jails across different regions of the world. These include North America, Central America, and the Caribbean [39], South America [40], Europe [41], Africa [42] and Asia [43]. Overall, the prevalence of COVID-19 is greater among prisoners compared to the general population. This is likely to be the case in Pakistan, which has one of the most overcrowded prison systems globally, with concerns about sanitation and available healthcare facilities [44], and at least 2313 prisoners testing positive for COVID-19 by August 2020 [45].

Blair et al. (2021) reported a 6% higher prevalence of COVID-19 in prisoners compared to the general population in Canada, and LeMasters et al. (2022) reported a 3% higher prevalence in the USA [46,47]. An earlier study from the USA also reported a higher mortality rate among prisoners compared to the general population [48].

Currently, more than 114 jails in Pakistan are accommodating 77,000 prisoners, which is beyond their capacity of 50,000, and the prevalence of COVID-19 among this population is currently unknown [44,49]. That being said, some reports claim that there were at least 2313 positive cases by August 2020 [45]. Since the start of the COVID-19 pandemic, more than 2400 prisoners in Pakistan have been diagnosed with chronic contagious diseases including hepatitis, HIV, and tuberculosis [50], and human rights groups have urged the authorities in Pakistan to take urgent measures for the protection of prisoners against COVID-19 [44,45,51].

In line with the recommendations of the World Health Organization (WHO), the government of Pakistan sought to ensure COVID-19 screening among prisoners, the provision of necessary personal protective equipment, and hygiene training, as well as to establish isolation centers for prisoners [50]. COVID-19 vaccine centers have also been established in district jails in Pakistan to vaccinate this highly vulnerable group. Moreover, the district administration in Punjab Province also tried to ensure prisoners were vaccinated with the first and second doses as well as booster doses. All prisoners would receive the COVID-19 vaccines free of charge, similar to all other groups in Pakistan.

In the initial phases of the vaccination campaign in Pakistan, HCWs who were assigned to jails were also vaccinated against COVID-19. In subsequent phases, other jail personnel were also vaccinated in order to reduce the threat of a COVID-19 outbreak in prisons. All visitors were also strictly asked to be vaccinated before entering the jails. To facilitate this, jail authorities established vaccination camps for the visitors to get vaccinated in case they could not be vaccinate earlier.

We are aware that hesitancy to receive the COVID-19 vaccines has been reported globally [52,53]. Vaccine hesitancy is due to many factors. These include concerns with the safety and efficacy of the vaccine, lack of trust in healthcare services during the pandemic, and lack of trust in health authorities as well as disbelieving the reality of COVID-19 [54,55,56,57,58]. Whilst various studies from Pakistan have shown vaccine hesitancy in the general population [59,60,61], HCWs [62,63], pregnant women [64], and hospitalized COVID-19 patients [65], as well as (surprisingly) among patients undergoing hemodialysis [25], we are currently unaware of any study that has been undertaken among prisoners in Pakistan regarding their COVID-19 vaccine status. In addition, currently there appears to be no research into the potential reasons for both acceptance and hesitancy towards COVID-19 vaccines among this target group during the ongoing mass-vaccination campaign throughout Pakistan. This is important given the vulnerability of this group, especially given current overcrowding in prisons in Pakistan, and concerns with hygiene and sanitation arrangements. Consequently, we sought to address this evidence gap by conducting a multicenter, cross-sectional study to determine the current COVID-19 vaccine status among prisoners in Pakistan as well as key factors associated with COVID-19 acceptance and hesitancy among prisoners. The findings can be used to guide future strategies to improve vaccination uptake in this vulnerable group if there are concerns with current hesitancy rates.

## 2. Materials and Methods

### 2.1. Study Design, Population, and Location

This descriptive, cross-sectional, questionnaire-based study was conducted among the prisoners of three district jails of Punjab Province in Pakistan over a period of four months (November 2022–February 2023). There are currently 36 district jails in Punjab (one in every district) besides the high security and central jails in the metropolitan cities [66]. All the district jails in Punjab have established vocational training centers as a part of the rehabilitation process where technical education and skills are imparted to the prisoners to facilitate their readjustment into society post discharge. Central jails in Punjab have factories and workshops which provide vocational training programs for the prisoners in different trades. These include carpet weaving, carpentry, dhurrie weaving, football stitching, blanket and bed sheet stitching for hospitals, furniture, and uniform stitching.

We contacted five district jail authorities through convenient sampling, and three of them were ready to allow data collection for this current project. We selected these five district jails initially based on the willingness of participation of HCWs assigned to these five jails.

Besides these, the health of prisoners is maintained by healthcare professionals (HCPs) including medical doctors, pharmacy/laboratory technicians, and other supporting staff present in prisons along with other HCPs who regularly visit the prisons to assist. All the prisoners in district jails have medical examinations and laboratory testing as well as medicines and vaccines available free of charge. Moreover, screening camps have also been arranged for all prisoners two to three times in a month where a team of HCPs from every specialty visit the facilities to help assess their health and provide care. However, there can be concerns with the level of care provided [44].

Since the start of the COVID-19 vaccine drive, prisoners have also been provided with vaccines free of charge if they wish to be vaccinated against COVID-19. Punjab Province was selected for this initial study due to the fact that Punjab contained ≥65% of the country’s population [67,68].

### 2.2. Study Instrument

The study instrument used in this survey was adapted from previous studies conducted among prisoners as well as other populations [59,60,61,63,64,65,69,70].

The subsequent content validity of the questionnaire was assessed by a six-member team. The team comprised medical doctors (MBBS degree holders) and pharmacists (PhD holders), as well as public health experts who were knowledgeable in this area. The panel thoroughly evaluated the questionnaire within a time period of a week and suggested minor revisions. After incorporating the recommendations and suggestions into the questionnaire, the final draft of the study questionnaire was prepared. A pilot study of the draft questionnaire was conducted among 20 potential participants prior to the initiation of the full-length study. These results were not included in the final study. The results of the pilot study were included in the questionnaire for use in the full study. After incorporating the recommendations of the pilot study participants, the study instrument contained the following four sections to evaluate the study outcomes (Appendix A):

**Appendix A** collected data on the demographics, health status, and COVID-19 vaccination status of the prisoners, e.g., their gender, age, education level, marital and posterity status, occupation before detention, activities in the prison, status of influenza and hepatitis B vaccines, and their current status of COVID-19 vaccination. There were no established age categories for the prisoners. These were decided following data collection to aid analysis.

**Appendix A** comprised nine questions relating to the factors associated with the acceptance of COVID-19 vaccines among the study population. Each question had ‘yes’ and ‘no’ options, and participants were requested to select one against each question.

**Appendix A** was designed to collect information about the willingness of the prisoners to be vaccinated against COVID-19 and what the potential facilitators of this were.

**Appendix A** collected information on the reasons for COVID-19 vaccine hesitancy among prisoners where this existed. There were 12 questions in this section. Each question had ‘yes’ and ‘no’ options, and participants were requested to select either ‘yes’ or ‘no’ against each question.

### 2.3. Sample Size Calculation

The sample size was calculated using the Raosoft sample size calculator 206-525-4025 (US). Assuming an expected frequency of 50%, a confidence interval of 95%, and a margin of error of 5%, the minimum sample size required for this study was 376 prisoners.

### 2.4. Data Collection Procedure

The team of investigators comprised HCPs who visited the respective district jails with the co-ordination of the HCPs assigned to these jails. The jail authorities were contacted and briefed about the study objectives and execution. The jail authorities were assured that the data were confidential and any participant could leave the data collection process at any stage without providing any reason, and this would not adversely affect their subsequent care. No personal information was obtained during data collection.

Written informed consent was obtained from every prisoner before being included in the study. After obtaining permission, data were collected from the prisoners with the help of jail staff who were trained by the investigators. We recruited study participants conveniently, and those were willing to participate were given the study questionnaire. Those who were unable to understand the questions were assisted by the prison HCWs who translated the questionnaire into the local language.

### 2.5. Ethical Considerations

The study protocol was approved by the Office of Research, Innovation and Commercialization, Lahore College for Women University. Furthermore, permission to conduct this study was also obtained from the concerned jail authorities.

### 2.6. Statistical Analysis

All data analysis was performed using SPSS version 22 for Microsoft Windows. Frequency and percentages were reported for categorical variables where mean ± standard deviation for continuous data. Demographic variables were compared between vaccinated and unvaccinated prisoners using the Chi-Square test. In addition, Phi and Cramer’s V were used to determine the strength of association between two nominal variables, where applicable. Statistical significance was taken as a *p*-value of less than 0.05.

## 3. Results

A total of 381 prisoners participated in the study out of 502 invited for participation, which provided a response rate of 79.5%. Demographic details of the inmates are presented in Table 1. The survey population was mostly between 18 and 44 years of age (64.3%), which was followed by those aged between 45 and 64 years (24.9%). There was a significantly higher proportion of unvaccinated prisoners in the younger age group (*p* < 0.001).

Around 85% of these were male, 62.2% resided in rural areas, 78.2% were unemployed before detention, and 81% shared a cell, with little difference between the groups. Regarding their education status, 33.1% were illiterate and 55.4% had a primary level education only, with little difference between vaccinated and unvaccinated individuals. None of the prisoners had received the influenza vaccine; however, 80.9% were vaccinated for hepatitis B with similar rates in both groups (Table 1).

A considerable proportion of inmates reported COVID-19 infections in their family members or relatives, and 32 (5.5%) reported having a family member and/or relative who succumbed to COVID-19. Again, these rates were similar among both vaccinated and unvaccinated inmates. Self-reported anxiety and depression levels were also similar between both groups.

As far as the COVID-19 immunization status of the prisoners was concerned, 53% of those vaccinated had received at least one dose of a COVID-19 vaccine (Figure 1). Of these (*N* = 202), the majority had two doses of a COVID-19 vaccine (*n* = 150) and 41 had received the additional booster.

The principal reasons for uptake of the vaccine among the vaccinated group included “fear of contracting SARS-CoV-2 infection” (56.9%), “desire to return to a pre-pandemic routine as soon as possible” (56.4%), “having no doubts on the safety of COVID-19 vaccines” (39.6%), and “I want to do my part in the fight against COVID-19” (35.6%) (Figure 2). 

As mentioned (Table 1), there was no statistically significant difference (*p* > 0.05) for any demographic variables between vaccinated and unvaccinated inmates except for age, which was strongly associated with COVID-19 vaccine uptake (χ^2^(3) = 76.645, *p* < 0.001, Cramer’s V = 0.457). The majority of the unvaccinated prisoners were between 18 and 44 years of age as compared to vaccinated inmates (85.5% vs. 45.5%), and most of the vaccinated prisoners were above the age of 45 as compared to unvaccinated prisoners (45–64 years old: 40.6% vs. 7.3%; >65 years old: 10.9% vs. 1.7%).

Of the 179 unvaccinated prisoners, only 16 (8.9%) showed intent or willingness to be vaccinated, with the remainder (*n* = 163, 91.1%) rejecting the COVID-19 vaccine. The top reasons for vaccine hesitancy among this groups are depicted in Figure 3. The principal reasons for vaccine hesitancy were: COVID-19 is not a real problem/disease (60.1%), safety concerns with the current COVID-19 vaccines (51.1%), the COVID-19 vaccine is a conspiracy (50.3%), “I don’t think I need this vaccine” (44.8%), and “I heard or read news that the COVID-19 vaccine is dangerous” (39.9%). Others reasons for COVID-19 vaccine hesitancy among prisoners were: if other people are having COVID-19 vaccines then I do not need it (33.7%) and concerns that COVID-19 vaccines will not work (33.7%). 

## 4. Discussion

We believe this is the first study conducted among one of the most neglected populations (prisoners) in Pakistan concerns the status of COVID-19 vaccines and factors associated with their acceptance or hesitancy in this population. COVID-19 vaccination is the key to controlling COVID-19 and its impact on healthcare systems globally, with many studies now showing the effectiveness of COVID-19 vaccines [22,23,71,72,73]. Vaccines have also been proven to lessen the transmission of many communicable diseases among prisoners [74], which is important for the COVID-19 vaccines. Considerable international legal legislation has compelled governments to adopt the principle of equivalence and to protect prisoners due to their higher vulnerability for a variety of diseases [75,76,77,78].

We observed that 53% of prisoners in our study had taken at least one dose of a COVID-19 vaccine during the COVID-19 vaccination drive in Pakistan, which included prisons. These findings are similar to previous studies from California where 49–56.2% of study populations were vaccinated against COVID-19 [69,79]. In contrast to our findings, a previous study from Zambia reported that 77% of prisoners were vaccinated against COVID-19 [80].

Among those who had been vaccinated against COVID-19 in our study, the majority (*n* = 150) received two doses and 11% received a booster dose as well. These findings were in contrast to the findings of a study from Italy where nearly half of the study participants had already received booster doses against COVID-19 [81]. It is thought that this will change in Pakistan as we are already seeing booster doses being rolled out, with a recent cross-sectional study in Pakistan suggesting that the prevalence of self-reported side effects with the COVID-19 vaccines were similar whether this was the first, second, or booster vaccination. In addition, most side effects were mild and transient, indicating the safety of the different COVID-19 vaccines [82]. As such, this helped to dispel some of the misinformation surrounding the COVID-19 vaccines.

The principal reasons for COVID-19 vaccine uptake among the prison population were “fear of getting COVID-19”, “desire to return to a pre-pandemic routine”, and “considering COVID-19 vaccine harmless/safe”. Others factors that prompted prisoners to get vaccinated included “participation in the fight against COVID-19” and “thought of being at high-risk of acquiring COVID-19”. This is similar to the findings of a previous study from Italy, where common reasons for COVID-19 vaccine acceptance among detained subjects were a higher risk of contracting COVID-19 as well as trust regarding the safety and availability of the COVID-19 vaccines [81].

Our study showed that the most common factors associated with COVID-19 vaccine hesitancy among prisoners were “misbelief regarding the real problem of COVID-19”, “concerned about the side effects of the COVID-19 vaccines”, and belief that “COVID-19 vaccine is a conspiracy”. There were similar concerns and issues among hemodialysis patients in Pakistan, another high-risk and vulnerable group [25]. Interestingly, there were no statistically significant differences in any of the studied demographic variables between vaccinated and unvaccinated prisoners except for age. In addition, there were no real differences in terms of the extent of family members contracting COVID-19 as well as dying from COVID-19. However, age was strongly associated with uptake and hesitancy. The majority of unvaccinated prisoners were aged between 18 and 44 years, with most of the vaccinated prisoners being above the age of 45. This may reflect the fact that the elderly were more likely to become seriously ill and die from COVID-19, certainly initially, with the young typically experiencing milder symptoms [19,22].

Previous studies have reported similar reasons for hesitancy including a fear of side effects and mistrust of health authorities [79,83,84,85]. In contrast to the findings of our study, another study conducted in Canada illustrated that the common reasons for COVID-19 vaccine hesitancy among prisoners were the “perception that the prisoners were at lower risk of catching the virus due to their restricted movement, poor health services, a feeling of being punished and failure to identify the benefits of COVID-19 vaccines” [86].

Prisoners are at high risk of contracting infections including COVID-19. Consequently, jail authorities should convey the possible threats to all prisoners about the risk of contracting COVID-19 due to overcrowding, challenges with minimum distancing requirements, as well as concerns with poor hygiene in prisons. Moreover, incarcerated people may access health experts and psychologists who can help them understand the benefits of vaccination by discussing the advantages of being vaccinated against COVID-19, along with addressing possible safety concerns [87]. In this respect, the recent cross-sectional study in Pakistan suggesting that the prevalence of self-reported side effects of the COVID-19 vaccines were similar whether this was the first, second, or booster dose, and that the side effects seen were mild and transient, should help address some of the safety concerns about the COVID-19 vaccines [82]. Family and friends are also important in this regard; consequently, jail authorities should engage with them to enhance vaccination rates among prisoners. This can be achieved by illustrating vaccine development, the current efficacy of the vaccines, and their magnitude as more data become available [60,87]. Local social influencers and religious leaders can also shift the attitudes of prisoners towards COVID-19 vaccines as well as help address some of the misconceptions about the vaccines promulgated via social media and other platforms [88,89]. Large-scale video messaging from imams of mosques is likely to also have positive spillover effects on swaying the prisoners’ behavior.

Although our study has highlighted a number of facilitators and barriers against vaccinations in a very marginalized population, we are aware that there are a number of limitations with our study. Firstly, this study was conducted in only three district jails of Punjab Province. However, we chose Punjab Province for this initial study for the reasons discussed in the methods section. Secondly, we are aware that we collected self-reported information from the prisoners that may be associated with reporting bias. However, we believe our approach is valid since it has been used by others. Thirdly, the status of the prisoners and their demographics, and the factors associated with either acceptance or hesitancy regarding the COVID-19 vaccines, were self-reported, which also carries the inherent bias associated with all self-reported data. Fourthly, the investigators were unable to evaluate the possible barriers faced by the prisoners in obtaining COVID-19 vaccines because of poor healthcare services that may have affected their COVID-19 vaccine status. Fifthly, we did not break the findings down into those who had already contracted COVID-19 and those who were virus-free; however, we did break the findings down into those who had family members who contracted and died from COVID-19. In addition, we did not assess the impact of the vaccines on reducing subsequent infection rates as this was not part of this research project. Lastly, as mentioned, the age categories of patients were developed as part of the data analysis to help with subsequent analyses. Despite these limitations, we believe our findings are robust and provide direction to the authorities in Pakistan as they seek to improve the vaccination status among this vulnerable group.

## 5. Conclusions

There is a concern regarding the considerable number of prisoners who are not vaccinated against COVID-19 in Pakistan since the initiation of COVID-19 vaccination drive. Most prisoners were hesitant due to conspiracy beliefs surrounding COVID-19 as well as safety concerns with the current vaccine. Health regulators and medical professionals need to work with jail authorities in Pakistan to address the highlighted concerns in order to maximize the coverage of the ongoing COVID-19 vaccination campaign among prisoners. In this way, the future health of this vulnerable group could be improved.

## Figures and Tables

**Figure 1 vaccines-11-01081-f001:**
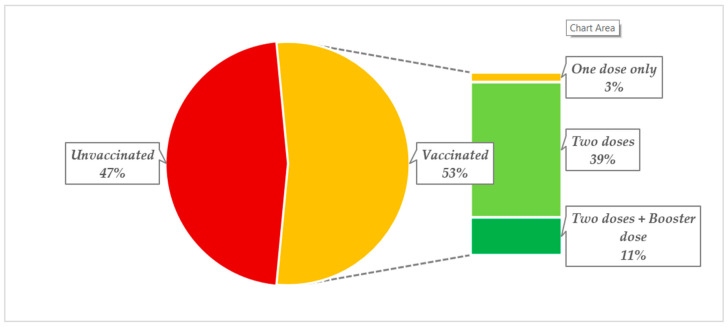
COVID-19 immunization status of incarcerated individuals.

**Figure 2 vaccines-11-01081-f002:**
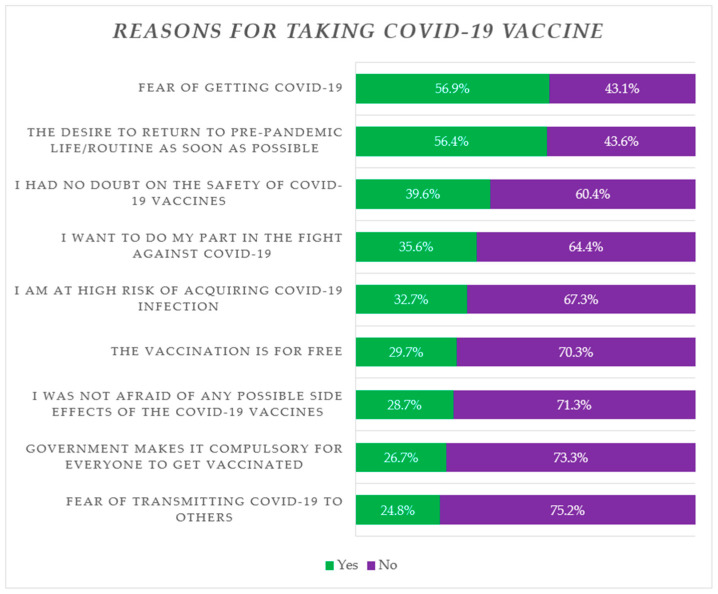
Reasons for vaccine acceptance ranked according to percentage of agreement (*N* = 202).

**Figure 3 vaccines-11-01081-f003:**
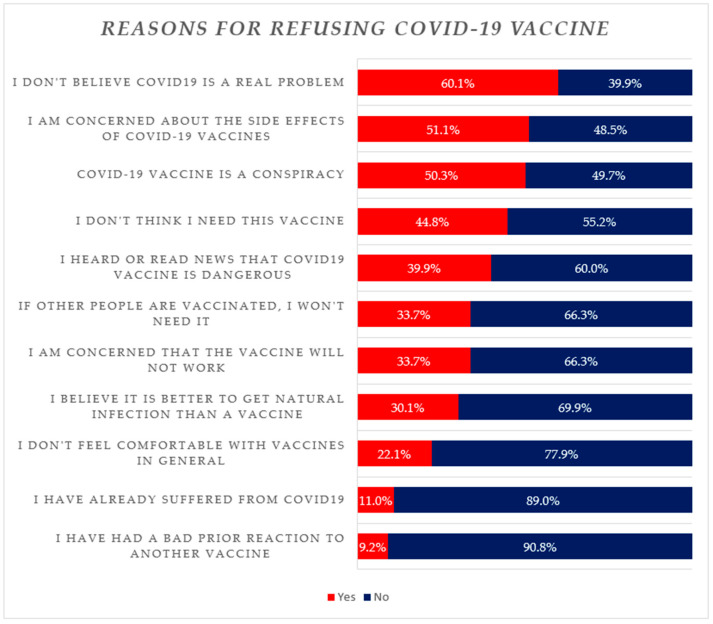
Reasons for COVID-19 vaccine refusal among inmates (*N* = 179).

**Table 1 vaccines-11-01081-t001:** Characteristics of COVID-19 for vaccinated vs. unvaccinated inmates.

Variable	Subgroups	*N* (%)			Sig.	Phi/Cramer’s V
		Overall (*N* = 381)	Vaccinated (*N* = 202)	Unvaccinated (*N* = 179)		
**Age (years)**	<18	16 (4.2)	6 (3.0)	10 (5.6)	**<0.001**	0.457
	18–44	245 (64.3)	92 (45.5)	153 (85.5)		
	45–64	95 (24.9)	82 (40.6)	13 (7.3)		
	≥65	25 (6.6)	22 (10.9)	3 (1.7)		
**Gender**	Male	324 (85.0)	174 (86.1)	150 (83.8)	0.523	0.033
	Female	57 (15.0)	28 (13.9)	29 (16.2)		
**Marital status**	Single	176 (46.2)	95 (47.0)	81 (45.3)	0.728	0.018
	Married	205 (53.8)	107 (53.0)	98 (54.7)		
**Education**	Illiterate	126 (33.1)	67 (33.2)	59 (33.0)	0.243	0.105
	Religious education only	32 (8.4)	12 (5.9)	20 (11.2)		
	Primary	211 (55.4)	115 (56.9)	96 (53.6)		
	Secondary or above	12 (3.1)	8 (4.0)	4 (2.2)		
**Occupation before detention**	Employed	83 (21.8)	49 (24.3)	34 (19.0)	0.214	0.064
	Unemployed	298 (78.2)	153 (75.7)	145 (81.0)		
**Housing**	Single cell	73 (19.2)	31 (15.3)	42 (23.5)	0.045	−0.103
	Shared cell	308 (80.8)	171 (84.7)	137 (76.5)		
**Influenza vaccine**	Vaccinated	-	-	-	-	-
	Not vaccinated	381 (100.0)	-	-		
**Hepatitis B vaccine**	Vaccinated	316 (82.9)	167 (82.7)	149 (83.2)	0.883	−0.008
	Not vaccinated	65 (17.7)	35 (17.3)	30 (16.8)		
**Family member or relative infected with COVID-19**	Yes	321 (84.3)	164 (81.2)	157 (87.7)	0.081	−0.089
	No	60 (15.7)	38 (18.8)	22 (12.3)		
**Family member or relative died due to COVID-19**	Yes	32 (8.4)	16 (7.9)	16 (8.9)	0.721	−0.018
	No	349 (91.6)	186 (92.1)	163 (91.1)		
**Self-reported anxiety and depression**	Yes	220 (57.7)	116 (57.4)	104 (58.1)	0.894	−0.007
	No	161 (42.3)	86 (42.6)	75 (41.9)		

## Data Availability

Further data is available from the corresponding author.

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
