# Peer review of "COVID-19 Vaccination Status as Well as Factors Associated with COVID-19 Vaccine Acceptance and Hesitancy among Prisoners and the Implications"

_vaccines, 2023, doi:10.3390/vaccines11061081_

Round 1

Reviewer 1 Report

This report could be improved by the following:

-The introduction should include the rationale for doing this study when prior studies of prisoners had already been published.

-This study should include comparable responses from similar aged non-imprisoned individuals from similar geographical populations. Perhaps even some historical or general data would suffice.

-The rationale for dividing age groups is unclear; was this done after the results were analyzed?

-The results of prisoners who already had COVID need to specifically presented and compared to those unvaccinated prisoners.

-The demographic information in the Abstract (lines 34-37) should be deleted.

Some moderate changes and emphases need to be made

Author Response

Reviewer 1

Open Review

Quality of English Language

( ) I am not qualified to assess the quality of English in this paper
( ) English very difficult to understand/incomprehensible
( ) Extensive editing of English language required
(x) Moderate editing of English language required
( ) Minor editing of English language required
( ) English language fine. No issues detected

Author response: Thank you for this. We have updated the manuscript with the help of one of the co-authors who is a native English speaker with over 500 papers published in peer-reviewed journals. We trust this is now OK.

Yes

Can be improved

Must be improved

Not applicable

Does the introduction provide sufficient background and include all relevant references?

( )

(x)

( )

( )

Are all the cited references relevant to the research?

(x)

( )

( )

( )

Is the research design appropriate?

( )

(x)

( )

( )

Are the methods adequately described?

(x)

( )

( )

( )

Are the results clearly presented?

(x)

( )

( )

( )

Are the conclusions supported by the results?

(x)

( )

( )

( )

Comments and Suggestions for Authors

This report could be improved by the following:

  1. The introduction should include the rationale for doing this study when prior studies of prisoners had already been published.

Author response: Thank you for the comment. Whilst prior studies have been reported among prisoners; only a limited number of studies have been reported from low-middle income countries (LIMICs) and no study to date from Pakistan. Consequently, we address this evidence gap by conducting the current study. We have also mentioned the same in introduction section, and trust this is now OK.

  1. Comment: This study should include comparable responses from similar aged non-imprisoned individuals from similar geographical populations. Perhaps even some historical or general data would suffice.

Author response: Thank you for this. We have upgraded the Introduction to include more data on vaccine hesitancy rates in LMICs in general and Pakistan in particular especially among adults, and trust this is now acceptable.

  1. The rationale for dividing age groups is unclear; was this done after the results were analyzed?

Author response: Thank you for the comment. The age groups were formulated after examing the data and age of study participants, and we have now mentioned this more in the Methodology section. This allowed us to present the data in uniform way in these groups, and we trust this is now acceptable.

  1. Comment: The results of prisoners who already had COVID need to specifically presented and compared to those unvaccinated prisoners.

Author response: Thank you for this comment. We did not specifically study this. However, as seen in Table 1 – the vast majority of prisoners (whether vaccinated or not) had family members with COVID-19 (higher number in the unvaccinated) and just under 10% had family members dying from COVID-19. We have now commented further on this in the Discussion and Limitations, and trust this is now OK

  1. The demographic information in the Abstract (lines 34-37) should be deleted.

Author response: Thank you. We have revised this section to only include key data. We trust this is now acceptable.

  1. Comments on the Quality of English Language. Some moderate changes and emphases need to be made

Author response: Thank you for this. We have updated the manuscript with the help of one of the co-authors who is a native English speaker with over 500 papers published in peer-reviewed journals. We trust this is now OK.

Reviewer 2 Report

Tittle: COVID-19 vaccination status, factors associated with COVID-19 vaccine acceptance and hesitancy among prisoners; findings 2 and the implication

-        In my opinion the title could be improved

Introduction and discussion

-        There were outbreaks of covid-19 among Pakistan prisons? Before and after vaccination?

-        Could authors comment how vaccination in all people working in prison was? and in visitors?

-        Did you study if influenza vaccines or others protect against covid-19?

-        How was the protection in people vaccinated?

-        Please correct, there are typo in some parts of the MS

Please consider to read the MS carefully and correct, some paragraph requires modifications

Please correct, there are typo in some parts of the MS

Author Response

Reviewer 2

Open Review

Quality of English Language

( ) I am not qualified to assess the quality of English in this paper
( ) English very difficult to understand/incomprehensible
( ) Extensive editing of English language required
( ) Moderate editing of English language required
(x) Minor editing of English language required
( ) English language fine. No issues detected

Author comments: Thank you for this. We have updated the manuscript with the help of one of the co-authors who is a native English speaker with over 500 papers published in peer-reviewed journals. We trust this is now OK.

Yes

Can be improved

Must be improved

Not applicable

Does the introduction provide sufficient background and include all relevant references?

( )

(x)

( )

( )

Are all the cited references relevant to the research?

(x)

( )

( )

( )

Is the research design appropriate?

(x)

( )

( )

( )

Are the methods adequately described?

(x)

( )

( )

( )

Are the results clearly presented?

(x)

( )

( )

( )

Are the conclusions supported by the results?

(x)

( )

( )

( )

Comments and Suggestions for Authors

1) Tittle: COVID-19 vaccination status, factors associated with COVID-19 vaccine acceptance and hesitancy among prisoners; findings 2 and the implication. In my opinion the title could be improved

Author response: Thank you. We have now amended the title and trust this is now OK.

2) Introduction and discussion - There were outbreaks of covid-19 among Pakistan prisons? Before and after vaccination?

Author response: Thank you. We have seen outbreaks before the vaccination programme – however, details are difficult to obtain (as now documented). There have been no recent figures. However, based on the findings from published papers – increasing vaccination rates should appreciably reduce the number of infections and the consequences. We trust this is now OK.

3) Could authors comment how vaccination in all people working in prison was? and in visitors?

Author response: As mentioned, in first phase of COVID-19 vaccination health care workers were vaccinated against COVID-19 including those that were deputed in the jails. In the subsequent phases all jail personnels were vaccinated. No visitor was allowed to visit the jail unless he/she was fully vaccinated. We have included this in the introduction section, and hope this is now OK.

4) Did you study if influenza vaccines or others protect against covid-19?

Author response:  Thank you for this comment. We didn’t study the effect of influenza or other vaccines against COVID-19. However – as stated – no study prisoner had received the influenza vaccine, however, 80.9% were vaccinated for hepatitis B (Table 1).

5) How was the protection in people vaccinated?

Author response: Thank you for this comment. We did not study the protection of vaccines against COVID-19 among prisoners as this was not the aim of this paper. We have though added this as a limitation and will be following up in future studies. We trust this is acceptable.

6) Please correct, there are typo in some parts of the MS

Author response: Thank you for this. We have updated the manuscript with the help of one of the co-authors who is a native English speaker with over 500 papers published in peer-reviewed journals. We trust this is now OK.

7) Comments on the Quality of English Language - Please consider to read the MS carefully and correct, some paragraph requires modifications

Author response: Thank you for this. We have updated the manuscript with the help of one of the co-authors who is a native English speaker with over 500 papers published in peer-reviewed journals. We trust this is now OK.

Round 2

Reviewer 1 Report

Thank you for your appropriate responses

Please have the grammar and contextual wording reviewed.